# Multifunctional Forestry and Interaction with Site Quality

**Ing-Marie Gren [1],\* and Franklin Amuakwa-Mensah [2]**

[1] Department of Economics, Swedish University of Agricultural Sciences, Box 7013, 75007 Uppsala, Sweden

[2] Environment for Development Initiative, University of Gothenburg, Box 645,
   SE 405 30 Gothenburg, Sweden, e-mail: franklin.amuakwa-mensah@efd.gu.se

\* Correspondence: Correspondence: ing-marie.gren@slu.se

**Abstract:** Several studies have shown the economic value of various ecosystem services provided by the forest. However, the economic value of how site-specific ecological conditions interact with other functions provided by the forest, such as timber value and carbon sequestration, has been less studied. As a result, this paper constructs a numerical discrete dynamic optimization model to estimate the economic value of site quality, taking into account its interaction with timber value and carbon sequestration, in Swedish forests. Analytical results show that the inclusion of the interaction of site quality with forest growth affects the optimal volume of harvest per year, compared to the case without consideration of site quality. The empirical results show that net present value, when considering timber values plus carbon sequestration and site quality interaction, is higher than the case where only timber and carbon sequestration were considered. However, the calculated net present value is sensitive to, in particular, the price of carbon sequestration and discount rate.

**Keywords:** timber benefit; ecological site quality; carbon sequestration; forest; Sweden

## 1. Introduction

Multifunctional forestry can be regarded as the ability of the forest to provide multiple and interconnected outputs or services, which can be either positive or negative, intended or unintended, complementary or substitute, and marketable or non-marketable [1]. Ecosystem services provided by the forest, besides the traditional timber provision includes recreation and ecotourism [2], erosion control from land-slides [3] and wind [2], flood protection and benefits to water quality, and climate regulation through carbon sequestration [4,5]. Furthermore, the presence of below-ground micro-organisms, mostly fungi diversity, plays an important role in enriching the site conditions of the forest as it helps in regulating ecosystem processes [6]. This role becomes apparent as below-ground organisms drive, or control, mineral and energy cycling within the ecosystems.

The productivity of the forest and the ability of the forest to provide essential ecosystem services are mostly driven by management practices and prevailing ecological quality at the site (site quality for short), which, in most cases are determined by environmental, biophysical, and climate conditions [7]. These site-specific ecological conditions have a tendency to interact with ecosystem services provided by the forest, which affects the economic value of the forest [8]. In spite of the potential benefits of site-specific ecological conditions in enhancing forest growth, there are no studies addressing the economic impacts of site quality on forest management. Therefore, the purpose of this study is to examine the economic effects of site quality in forest management when including both timber values and non-market services, in terms of carbon sequestration. This objective is achieved by constructing a discrete dynamic optimization problem, where the forest manager maximizes his/her net present values (NPV) of both timber and carbon sequestration subjected to the development in the standing biomass volume, which is dependent on site quality.

Following the work by Gren and Amuakwa [9] and earlier studies [10,11], we used site index as a proxy for site productivity or quality. According to Skovsgaard and Vanclay [10], site index shows the potential of tree growth under ideal conditions, and is usually measured as the biomass potential at a certain age of a tree species. Moreover, Stokland et al. [12] asserts that site index usually reflects the quality of the forest soil in terms of potential forest productivity. Our evaluation of site quality is analyzed by considering the effect of site quality on forest growth rate.

The present study is similar to the literature on the economics of management of multifunctional forests, which can be traced to the novel contribution by Hartman [13]. Several studies have added carbon sequestration to timber values [9,13–23]. The studies differ with respect to the choice of forest growth functions, inclusion of carbon sequestration, and empirical application. A common result is that the rotation period increases when including carbon sequestration, in particular when the discount rate is low.

To the best of our knowledge, no previous study has examined the interaction between site quality and the provision of timber outputs and carbon sequestration. The novelty of this study is thus the consideration of this interaction where site quality affects harvest, and in turn has impacts on site quality. The remainder of the study is organized as follows; section two presents the structure of the numerical dynamic optimization model used in the study. Data retrieval is presented in section three. Section four presents and discusses the results from the study, and the final section provides concluding remarks.

## 2.A Model of Site Quality and Management of Multifunctional Forests

The optimal forest management in the presence of site-specific ecological conditions and the economic evaluation of these in Swedish forest is calculated based on a non-linear discrete-time dynamic programming model. The level of site quality in time $t + 1$ is assumed to be determined by site quality in the earlier period, and by harvest. Harvest creates disturbance in the soil, mainly in terms of carbon losses, which affects the level of the quality [24,25]. As shown by Diochon et al. [25], carbon losses can occur decades after the harvesting of spruce, but can also recover after an even longer period due to the carbon content in intact forests. However, there is insufficient information on the impact of harvest on the more general site quality indicator used in this study, which includes multiple environmental factors at the site (see Section 3). Therefore, a simple linear relation was introduced between harvest in time $t$, $H_t$, and site quality in the subsequent period, $D_{t+1}$. Such a simplification was also made for natural recovery of the soil. Statistical analysis of soil quality and harvest for Swedish forests by Amuakwa [26] indicates that such a linear specification of the soil quality dynamics gives a good statistical fit to the data. The dynamics of site quality is then written as:

$$D_{t+1} = \eta_1 D_t - \eta_2 H_t, \qquad D_0 = \bar{D}_0 \qquad (1)$$

where $\eta_1 > 1$ is the natural recovery, and $\eta_2$ is the impact of harvest on forest growth.

The development of site quality is assumed to affect the standing forest volume proportionally [9]. The change in standing biomass volume between two periods is then determined by forest growth $G(B_t)$, harvest $H_t$ and site quality $D_t$. Harvest is assumed to take place at the end of each year. Formally, we express the change in standing forest biomass volume as:

$$B_{t+1} - B_t = G(B_t) - H_t + \tau D_t, \qquad B_0 = \bar{B} \qquad (2)$$

where $B_{t+1}$ and $B_t$ are the levels of standing biomass volume in the periods $t + 1$ and $t$ respectively, and $\tau$ is the proportional effect of site quality on standing biomass growth.

In the present study, only above ground carbon sequestration is included, which depends on the forest biomass growth described in Equation (2). In addition, the net removal depends on the use of forest products. Following the work of Vass and Elofsson [21], we assume that forest products in the

process of decay releases carbon into the atmosphere over time. The expression of net carbon dioxide removal by forest products is given as:

$$F(H_t, H_{t-1}, \dots, H_0) = \omega H_t - \varphi H_t - \theta \left[ \sum_{v=0}^{t-1} (1 - \theta)^{t-1-v} - H_v \right] \qquad (3)$$

In Equation (3), the first term represents the carbon content stored in harvested forest products, where $\omega$ is a conversion factor that converts cubic meters of wood volume to metric tons of carbon (that is, the carbon content of wood ($CO_2/m^3$). The second term, $\varphi H_t$, is the carbon emissions resulting from harvesting, transporting, and processing of biomass. The third term in Equation (3) is the release of carbon from the stock built up in foregoing periods "$v$", due to the decay of wood. Thus, this term captures the carbon released into the atmosphere as the wood decays over time.

Based on the two main components of net carbon sequestration, the value of carbon sequestration in monetary terms can be expressed as

$$Card_t = P_t^{co} \left[ \omega(B_{t+1} - B_t) + F(H_t, H_{t-1}, \dots, H_0) \right] \qquad (4)$$

where $P_t^{co}$ in Equation (4) is the unit value of carbon sequestration.

With respect to timber benefit, the forest manager receives revenue from the sale of forest products and also takes into consideration the cost of production, which includes annualized capital cost and harvest cost. Thus, the timber benefit can be expressed as:

$$TimBen_t = P_t^{Tim} H_t - Cost(H_t) \qquad (5)$$

Where, in Equation (5), $P_t^{Tim}$ is the market price for timber and $Cost(H_t)$ is the cost function of timber production.

The decision problem is formulated as the allocation of harvest over time, $H_t$, to maximize NPV of current and future streams of benefits from carbon sequestration and timber values, $\pi$, during a certain time frame, subjected to the development in the standing volume of biomass and the dynamics of site quality, which is formulated as:

$$\underset{H_t}{Max}\,\pi = \sum_{t=0}^{T} \rho^t \{ P_t^{Tim} H_t - Cost(H_t) + P_t^{co}[\omega(B_{t+1} - B_t) + F(H_t, H_{t-1}, \dots, H_0)] \} \qquad (6)$$

Subject to Equation (1) and (2).

where $\rho^t = \left( \frac{1}{1+\delta} \right)^t$ is the discount factor, with $\delta \geq 0$ (i.e., the discount rate). This implies that future profits are of less value to the forest manager than current profits.

As shown in Appendix A, the condition for an optimal solution to the decision problem in Equation (6) is that the marginal benefit and marginal costs of harvest are equal in each period of time, which is expressed as;

$$P_t^{Tim} + P_t^{co} \left[ (\omega - \varphi) - \sum_{v=t+1}^{T} \rho^{v-t} \theta (1 - \theta)^{t-1-v} \right] = \frac{\partial Cost(H_t)}{\partial H_t} + \rho(\lambda_{t+1} + \eta_2 \mu_{t+1}) \qquad (7)$$

The terms on the left hand side of Equation (7) show the marginal benefits; and the terms on the right side, the marginal costs. The first term on the left hand side is the marginal benefit from timber sale (that is, the unit price of timber), and the second term is the marginal benefit from carbon sequestration. It is comprised of the product of carbon price and the marginal net carbon dioxide

removal by forest products (that is, $(\omega - \varphi) - \sum_{v=t+1}^{T} \rho^{v-t} \; \theta \; (1 - \theta)^{t-1-v}$). The marginal net carbon dioxide removal by forest products is comprised of the reduction in carbon added to the forest product stock (that is, $(\omega - \varphi)$, and the avoided release of carbon in the future period when less wood is used for forest products at time $t$ (that is, $\sum_{v=t+1}^{T} \rho^{v-t} \; \theta \; (1 - \theta)^{t-1-v}$).

The first term on the right hand side of Equation (7) is the marginal cost of harvest, and the second term shows the impact on future discounted profits from a harvest by one unit in period $t$. Future impacts occur through the effects of forest biomass and site quality growth. On the right hand side of Equation (7), $\lambda_{t+1}$ is the marginal user cost of biomass (also denoted as the shadow price of biomass) which shows the value of the impact on total future net benefit on standing forest in period $t + 1$ for each additional unit of harvest in period $t$. In other words, the marginal user cost or shadow price corresponds to foregone future profits per unit of biomass that is used today. Also, $\mu_{t+1}$ is the shadow price of site quality, and it shows the value of the impact on the forest manager's total future net benefit via biomass volume for each additional unit of site quality.

Depending on the impact of harvesting on standing biomass growth and carbon sequestration, the marginal user cost can either be positive or negative. In situations where the average forest stand is young with relatively low growth, for each additional unit of standing volume harvested today, there is an associated cost. This is because the average age and standing biomass volume falls in the next period, which implies a lower growth in standing volume, given that the average forest stand is young. As a result, marginal user cost is positive. On the other hand, in situations where the average forest stand is old with relatively low growth, for each additional unit of standing volume harvested today, there is an associated benefit. This is because the average forest age falls in the next period, which implies a higher growth in standing volume, given that the average forest stand is old. In this case, the marginal user cost is negative.

Assuming that the marginal user cost of forest biomass is positive, Equation (7) shows that neglecting any of the future impact on forest biomass or site quality will result in too high harvest volumes. In this case, the total marginal cost is reduced, which implies higher harvest in early periods. This, in turn, implies lower profit during the entire period.

## 3. Description of Data

In order to calculate the role of site quality, data are needed for parameterizing the growth and other functions presented in Section 2. The analysis considers the entire productive forest of Sweden without particular emphasis on tree species since there is no specific biomass growth function for various tree species.

We use a site index for Swedish forests which is based on statistical assessments of multiple effects of different environmental factors at a site [27]. According to Bontemps and Bouriaud [11], the index can indicate the constraints of the ecological niche and distribution of tree species, and thus can portray biodiversity. As suggested by Stokland [12], site quality can be an indicator of this type of diversity, since it usually reflects the quality of the forest soil in terms of potential forest productivity. It has also been found to be correlated with fungi diversity in forest soil in Sweden [28].

Parameter values of $\eta_1$ and $\eta_2$ in the dynamics of site quality in Equation (1) are obtained from Amuakwa-Mensah [26], who performed an econometric analysis with annual data on site quality and harvest over the period 1965–2013. Data were obtained from the Swedish Forest Agency [29] that measures and reports site quality index for different parts of Sweden, the calculation of which is based on Hägglund and Lundmark [27]. The value of the index ranges between 2.6 and 8.9 for different time periods and forest regions in Sweden, with the highest in the regions in the south of Sweden. Because of the time series nature of the harvest and site quality data, fully modified ordinary least squares estimation techniques were used in order to address the problems of serial correlation, stationarity, and endogeneity. The estimated parameter values of $\eta_1$ and $\eta_2$ are presented in Table 1.

The standing biomass growth $G(B_t)$ in Equation (2) assumes the form of a logistic growth function. Using annual forestry data for about 50 years, Gren and Amuakwa-Mensah [9] estimated a logistic growth function for Swedish forests, which is given as:

$$G(B_t) = \gamma_1 (1 - \frac{B_t}{\gamma_2}) B_t \tag{8}$$

Where $\gamma_1$ and $\gamma_2$ in Equation (8) are intrinsic growth rate and carrying capacity, respectively. The values for these parameters are shown in Table 1. The marginal effect of site quality on biomass growth ($\tau$) in Equation (2) is obtained from Gren and Amuakwa-Mensah [9].

With respect to the net carbon dioxide removal by forest products function in Equation (3), the parameter $\varphi$, which is the fraction of harvested biomass lost, is obtained from Vass and Elofsson [21]. The parameter is obtained by dividing the emissions from harvesting, transporting, and processing of bioenergy, measured in tonnes $CO_2/m^3$, by the carbon content of wood. For countries in the boreal forest region, the value of this parameter is about 0.026. In relation to the rate of decay, $\theta$, of forest products, we rely on the work by Stainback and Alavlapati [16]. Since our study focuses on an aggregated analysis, we constructed a composite index for the decay rate of forest product by considering saw-timber and pulpwood [28]. This implies that the relative share of saw-timber and pulpwood in our analysis is about 0.52 and 0.48, respectively [28]. We obtained the weighted average of the rate of decay of forest products by multiplying the shares of saw-timber and pulpwood with their respective decay rates and summing them up. This gives us a weighted average rate of decay, $\theta$, of 0.416. The carbon content per $m^3$ wood, $\alpha$, is obtained from Vass and Elofsson [21], and the parameter used in this study is for countries with boreal forests, since Sweden is within that category.

Given that the annual average price of saw-logs and pulpwood in Sweden for the year 2015 is 504 SEK/$m^3$ fub (solid volume excluding bark) and 277 SEK/$m^3$ fub, respectively, we also calculated the weighted price of timber to be 395 SEK/$m^3$ fub (9.36 SEK = 1 Euro on average in 2015). We used the Swedish carbon tax of about 1130 SEK/ton as carbon price in our analysis [30]. The study uses a discount rate of 3%, which is similar to that of Vass and Elofsson [21] and falls between the rates applied by Stern [31] and Nordhaus [32]. Also, the discount rate in this study is equivalent to a fifteen year average (that is, between 2000 and 2015) of the Swedish government 5-years bond. According to Boardman et al. [33], a discount rate of 3% is reasonable for public projects.

A quadratic cost function was assumed where cost varies with harvest, which is defined as:

$$C(H) = \beta_0 + \beta_1 H + \beta_2 H^2 \tag{9}$$

Parameter values of the quadratic harvest cost function in Equation (9) are found in Gren and Amuakwa-Mensah [26], which made econometric estimates based on cost estimates by Lundmark et al. [34]. For a list of all parameter values and sources, see Table 1.

**Table 1.** Functions, parameter values and data sources.

| Function/Parameter | Data source | Baseline values |
|:---:|:---:|:---:|
| $\delta$ | Sveriges Riksbank [35] | 3% |
| $P^{Tim}$ (2015) | Swedish Forest Agency [29] | 395 SEK/$m^3$ fub |
| $P^{CO}$(2015) | Fores [30] | 1130 SEK/ton |
| $\tau$ | Gren and Amuakwa-Mensah [9] | 0.023 |
| $\varphi$ | Vass and Elofsson [21] | 0.026 tonne $CO_2/m^3$ |
| $\theta$ | Stainback and Alavalapati, [16] | 0.416 tonne $CO_2/m^3$ |
| $\alpha$ | Vass and Elofsson [21] | 0.912 tonne $CO_2/m^3$ |
| Initial biomass $(B_{2015})$ | Swedish Forest Agency [29] | 138 $m^3$ sk/ha [1] |
| $\gamma_1$ | Amuakwa-Mensah and Gren [9] | 0.132 |
| $\gamma_2$ | Amuakwa-Mensah and Gren [9] | 628. $m^3$ sk/ha |

| $\eta_1$ | Amuakwa-Mensah [26] | 1.0064 |
| $\eta_2$ | Amuakwa-Mensah [26] | 0.0007 |
| Cost function | Amuakwa-Mensah [26] | $\beta_0 = 749.9$, $\beta_1 = 65.4$, $\beta_2 = 1.66$ |

[1] m³ sk/ha = cubic metre standing volume overbark per hectare.

## 4. Results and Discussion

Based on the parameter values from the biological and economic models shown in Table 1, the optimal time path for harvesting, carbon benefit, profit, and the economic value of site quality is generated through simulation for a period of 100 years. The numerical models are solved using the mathematical programming code in GAMS [36], using the year 2015 as the baseline period and making simulations for a 100 year period. A stepwise method was followed in this analysis, in order to ascertain the economic value of various ecosystem services from the forest. First, the optimization process was made, considering only timber output of the forest. In the second step, optimization was made over timber and site quality interaction; and the third step considered both timber and carbon values, without site quality interaction. The final step considers timber and carbon values together, with the interaction of site quality.

In order to evaluate the impact of different parameter values on the maximum NPV, sensitivity analyses were performed where the levels of the discount rate, impact of site quality on biomass growth rate, and intrinsic growth rate of biomass were changed.

### 4.1. Base Case

The NPV of forest management under the four different cases (timber with and without carbon sequestration, and with and without site quality interaction) is represented in Figure 1. As expected, accounting for the non-timber value of the forest increases the NPV.

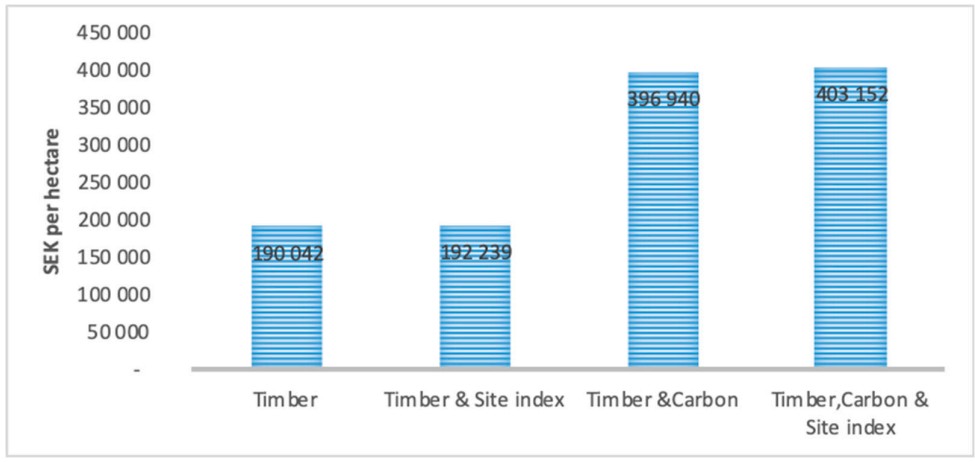

**Figure 1.** NPV of Swedish forest management under different cases in SEK[a] per ha; [a] (9.36 SEK = 1 Euro on average, in 2015).

The results show that the NPV of forest management more than doubled when considering carbon sequestration; it increased from 190,042 SEK per hectare to 396,940 SEK per hectare. The impact of site quality was more modest; where the NPV for timber and site quality interaction, and timber, carbon sequestration, and site quality interaction are 192,239 SEK per hectare and 403,152 SEK per hectare, respectively. The value of site quality can then be calculated as the difference in NPV with and without the consideration of site quality in the optimization, which gives 2197 SEK/ha with timber as output, and 6212 SEK/ha with timber and carbon sequestration.

There are small differences in the optimal time path of harvest between the scenarios. In all cases, the harvest path shows a similar trend where it rises during initial years, remains relatively constant

over a long period, and rises sharply during the later years (Figure A1 in Appendix B). Comparing the harvest path for the case where only timber value is optimized with that of timber and carbon sequestration, the volume of harvest when only timber is optimized is higher at the initial years until the end of the period. This finding corroborates that of [37], where it was observed that accounting for carbon value in the optimization process reduces the volume of harvest at initial years. Similarly, [16], [15], and [20] also conclude that the inclusion of carbon sequestration and other ecosystem benefits of the forest in the optimization process results in longer rotation age.

*4.2. Sensitivity Analysis*

The results in the base case indicated relatively small effects on the maximum NPV of the impact of site quality on forest growth. Therefore, calculations were made for a high increase in this parameter value, at 0.5 instead of 0.023 in the base, to examine the impact on NPV. Similarly, calculations are made for a relatively large decrease in the price of carbon sequestration. In the base case, the Swedish tax on carbon dioxide emissions (SEK 1130/ton $CO_2$) was used, which is a relatively high value of the social cost of carbon [31,32]. Therefore, calculations were made using the equilibrating price at the EU emission trading system (ETS) as the price of carbon, which was SEK 73/ton $CO_2$ in 2015. Impacts on NPV were also calculated for large changes in the discount rate, from 3% in the base case, to 1% and 6%. On the other hand, calculations were made for a relatively small change in the intrinsic forest growth rate, at 0.16 instead of 0.13 in the base case. Despite the large increase in the impact of site quality on forest growth, the effects on maximum NPV is in the same order of magnitude as the increase in the forest growth rate (Table 2).

**Table 2.** Comparison of NPV between baseline and different values of site quality impact, carbon price, intrinsic growth rate, and discount rate, in thousand SEK/ha.

| | Baseline | High site quality impact | Low price of carbon | High intrinsic growth rate | Discount rate: | |
| --- | --- | --- | --- | --- | --- | --- |
| | | | | | 1% | 6% |
| Timber only | 190 | | | 241 | 484 | 67 |
| Timber and site quality | 192 | 238 | | 243 | 489 | 68 |
| Timber and carbon | 397 | | 250 | 451 | 600 | 273 |
| Timber, carbon, and site quality | 403 | 454 | 252 | 457 | 626 | 275 |

From Table 2, the NPV for timber, and both timber and carbon interactions, with high impact of site quality is 238 thousand SEK per hectare and 454 thousand SEK per hectare, respectively. The corresponding site quality values are then 48 thousand SEK per hectare and 57 thousand SEK per hectare for interactions with timber only, and both timber and carbon, respectively. A high intrinsic growth rate allows for higher harvest levels and carbon sequestration, and NPV increases for all combinations of the ecosystem services. The values of site quality are in the same order of magnitude as the case with high impact of site quality on forest growth.

As expected, NPV is sensitive to the price of carbon and discount rate. A decrease in the price of carbon from the Swedish tax to the EU ETS price of carbon reduces the NPV by almost 40%. An increase in the discount rate to 6% leads to a drastic decrease in the NPV. In relation to site quality value, we observe from Figure A2 in Appendix B, an inverse relationship between site quality and the discount rate. Thus, at a lower discount rate, site quality value is higher than at a higher discount rate. The calculated NPV is also sensitive to the assumed values of timber prices, but the magnitude of the impacts of these parameters is smaller than that for the parameters included in the present study [26].

## 5. Conclusions

Given the multifunctionality of forests in providing both timber and non-timber services, several studies have attempted to quantify the economic value of non-timber services provided by forests. However, no study has examined implications of the interactions of the services provided by the forest with site specific ecological conditions. As such, using a numerical discrete dynamic optimization model, we estimated the economic value of site quality, taking into account that harvest has a negative impact on forest biomass and site quality growth. The contribution of site quality becomes apparent as it facilitates biomass growth. With high growth, there is an increase in timber yield, which is eventually sold for more net revenue. In addition, higher biomass growth increases the carbon sequestration potential of standing biomass, and this increases the non-timber benefits of the forest to the manager. However, the interaction also implies that the marginal cost of harvest increases, due to the negative effect on site quality and associated effects on future biomass growth.

The empirical application to the forests in Sweden showed that the value of site quality ranges between 2197 SEK per hectare and 6212 SEK per hectare depending on assumed interactions with timber and carbon sequestration. Given that the productive forest area in Sweden is about 22.7 million hectares, the value of site-specific ecological conditions in Swedish forests ranges between approximately SEK 50 billion and SEK 141 billion, which correspond to approximately 0.1%–0.2% of the gross domestic product in 2015. The value increases considerably when the impact of site quality on forest growth increases.

However, the empirical calculations rest on simplified assumptions, in particular, on the impacts of harvest on site quality and on the effect of site quality on forest growth. This points to the need for more empirical analyses of these relations. The empirical results also showed that the NPV of forest management can be doubled when considering the value of carbon sequestration, which is demonstrated in several other studies. The results therefore indicate a need for careful analyses of the economic role of site quality and carbon sequestration of Swedish forests. A materialization of these values would require the implementation of an appropriate incentive structure for the forest managers. The approach used in this presented study could then be a useful tool for deriving the effects of different incentive structures.

**Author Contributions:** conceptualization Ing-Marie Gren and Franklin Amuakwa-Mensah.; methodology Ing-Marie Gren, X.X.; software, Ing-Marie Gren and Franklin Amuakwa-Mensah.; formal analysis, Ing-Marie Gren and Franklin Amuakwa-Mensah.; investigation,Ing-Marie Gren.; data curation, Franklin Amuakwa-Mensah.; writing—original draft preparation, Franklin Amuakwa-Mensah.; writing—review and editing, Ing-Marie Gren.; supervision,Ing-Marie Gren; project administration, Ing-Marie Gren.; funding acquisition, Ing-Marie Gren. All authors have read and agreed to the published version of the manuscript.

**Funding:** Funding was provided by the Swedish research council Formas for the research project 'The roles of fungal diversity, forest management and their interactions - analysis and valuation of effects on ecosystem services' (Grant no. 215-2012-1257).

**Acknowledgments:** We are much indebted to two anonymous reviewers for useful comments.

**Conflicts of Interest:** The authors declare no conflict of interest.

## Appendix A: **Derivation of conditions for optimal harvest**

In order to the solve the decision problem described in Equation (6), we form the discrete time Lagrangian problem as:

$$
\begin{aligned}
L = \sum_{t=0}^{T} \rho^t \Big\{ & P_t^{Tim} H_t - Cost_t(H_t) + P_t^{CO}[\omega(B_{t+1} - B_t) + F(H_t, H_{t-1}, \dots, H_0)] \\
& + \rho\lambda_{t+1}[B_t - B_{t+1} + G(B_t) - H_t + \tau D_t] \\
& + \rho\mu_{t+1}[\eta_1 D_t - \eta_2 H_t - D_{t+1}] \Big\}
\end{aligned}
\tag{A1}
$$

By assuming an interior solution, we derive the necessary first-order conditions for profit maximization, to obtain the optimal values of $H_t$ and $B_t$. The first-order conditions are shown below:

$$\frac{\partial L}{\rho^t \partial H_t} = P_t^{Tim} - \frac{\partial Cost(H_t)}{\partial H_t} + P_t^{co}\left[(\omega - \varphi) - \sum_{v=t+1}^{T} \rho^{v-t}\,\theta\,(1-\theta)^{v-1-t}\right] - \rho(\lambda_{t+1}$$
$$+ \eta_2\mu_{t+1}) = 0 \tag{A2}$$

$$\frac{\partial L}{\rho^t \partial B_t} = P_t^{co}\omega\left(\frac{1}{\rho} - 1\right) + \rho\lambda_{t+1}\left(1 + \frac{\partial G}{\partial B_t}\right) - \lambda_t = 0 \tag{A3}$$

$$\frac{\partial L}{\rho^t \partial D_t} = \rho\left(\tau\lambda_{t+1} + \eta_1\mu_{t+1}\right) - \mu_t = 0 \tag{A4}$$

$$\frac{\partial L}{\rho^t \partial \rho\lambda_{t+1}} = B_t - B_{t+1} + G(B_t) - H_t + \tau D_t \tag{A5}$$

$$\frac{\partial L}{\rho^t \partial \rho\mu_{t+1}} = D_t + R(D_t) - \eta H_t + D_{t+1} = 0 \tag{A6}$$

**Appendix B: Figures A1–A2**

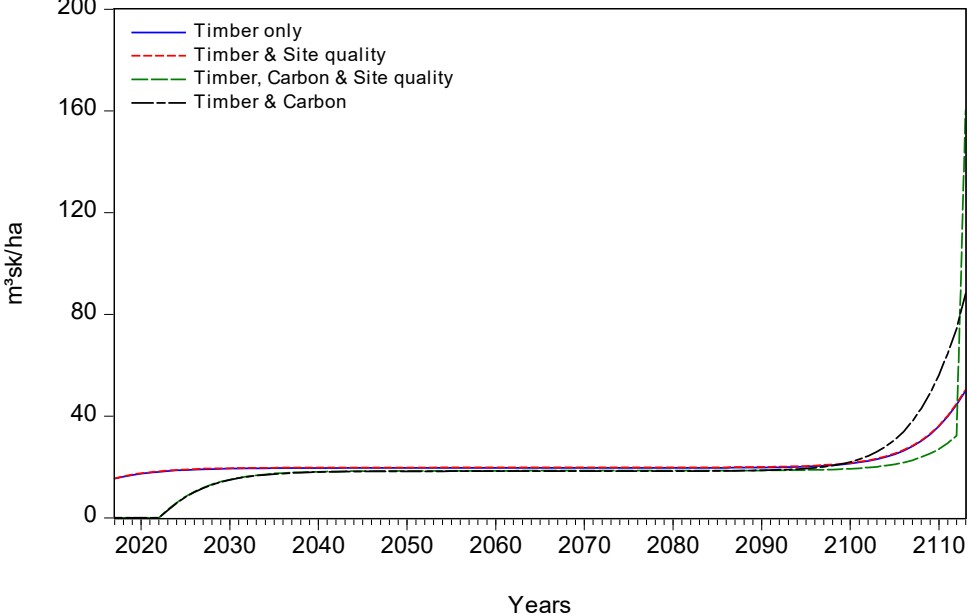

**Figure A1.** Optimal time path of harvest.

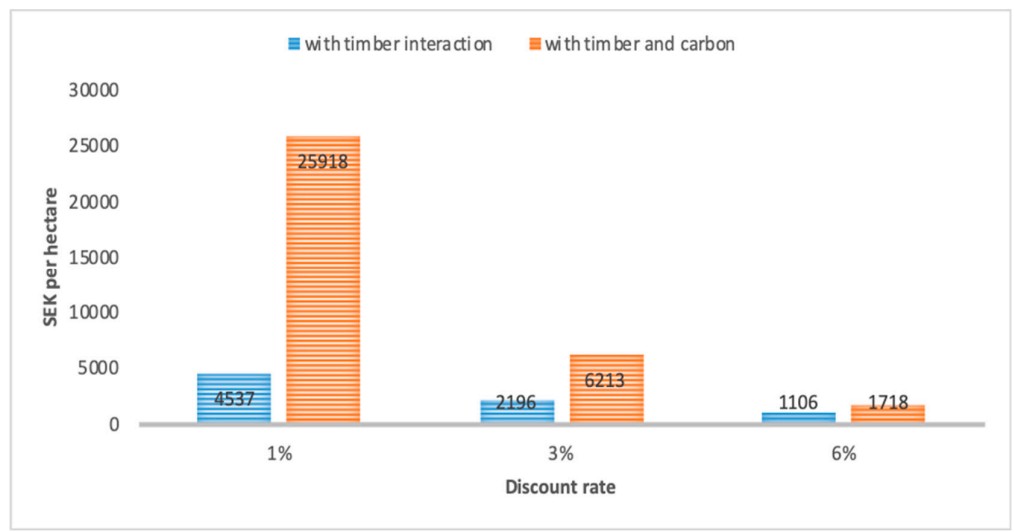

**Figure A2.** Value of site quality for different discount rates.

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
