# Peer review of "Multifunctional Forestry and Interaction with Site Quality"

_forests, doi:10.3390/f11010029_

Round 1

Reviewer 1 Report

The manuscript addresses an important aspects - interaction of site quality with timber production and carbon sequestration. There is no doubt about the impact of site quality on timber production and carbon sequestration, but the manuscript has some issues that need to be addressed.

Equation 1: The authors have conceptualized the positive contribution of the growth function and the negative contribution of timber harvest on site quality.  On one side, timber harvesting may lead to disturbances which may have negative impact on site quality but timber harvesting may also leave some litter (roots, leaves, twigs etc) in the area that may enrich the soil. In short, it is important that the authors include some details about their assumptions, including linear relationship between the site quality and harvest. Equation 3 is not clear. The authors state that Ht represents the carbon stored in the timber harvested. So is Ht carbon or timber volume. If it is carbon, it should be represented by some other symbol because Ht is used for timber in equations 1 and 2. Please clarify this equation. If you are going to use eq (8) for your estimation of site quality, why to provide Eq (1). Or identify the use of eq (8) as one of your limitations in your conclusion section. What is the unit of site quality? In sensitivity analysis, the authors state that by increasing the impact of site quality from 0.023 to 0.5, what is the unit of your impact? Why such a big increase from 0.023 to 0.5? Is such a big change possible? This choice does not seem reasonable. How do you define high site quality? What is the actual range of site quality in Sweden? Line 254 and 255: Site quality values are stated. I am not sure what do you mean by site quality value (SEK/hectare). Table 1: Sources - Please include citation numbers that you used in your reference section. I could not find some of your citations in this table in your reference section.     

Author Response

See enclosed file.

Reviewer 2 Report

This study examined the economic effects of site quality in forest management when including both timber values and non-market services in terms of carbon sequestration by the discrete dynamic optimization model with four scenario: timber, timber and site quality, timber and carbon sequestration, and all three included. The model employed is sensible and the results can appropriately explain the research issue. However, some parts need improvement.

The model applied in this study ignored the elements of uncertainty such as prices of timber and carbon. To test the uncertainty of the model, they conduct a sensitivity analysis by changing three parameters’ value: site quality, growth rate, and discount rate. Did this test conducted by changing the value once? Or conducted simulation for thousands times? For example, the value of site quality changed from 0.023 to 0.5. Did you randomly select the value from that range and conduct the simulation simultaneously? Please clarify how you made the sensitivity analysis.

Besides, some minor revisions are needed.

Line 223: “The net present value (NPV) of forest management under the four different cases; timber with and without carbon sequestration and with and without site quality interaction is represented in Figure 1.” Abbreviation should be used at the first time.

Line 23: “In addition, the forest provides habitat for fungi species some of which are used for food or traditional medicines (Nghiem, 2014).” The format of reference cited should be consistent, either number or name and year.

Line 23/45/71/72/75/139/172/256: have extra space between words. Please double check and remove extra space.

Table 2: the capital of the first column is not consistent.

Author Response

See enclosed file.

Round 2

Reviewer 1 Report

The authors have addressed my suggestions/observations.

Reviewer 2 Report

Can be accepted after proofreading.